# Nanocapsule of MnS Nanopolyhedron Core@CoS Nanoparticle/Carbon Shell@Pure Carbon Shell as Anode Material for High-Performance Lithium Storage

**DOI:** 10.3390/molecules28020898

**Published:** 2023-01-16

**Authors:** Peng Yang, Yongfeng Yuan, Dong Zhang, Qiuhe Yang, Shaoyi Guo, Jipeng Cheng

**Affiliations:** 1College of Machinery Engineering, Zhejiang Sci-Tech University, Hangzhou 310018, China; 2Hang Zhou City of Quality and Technical Supervision and Testing Institute, Hangzhou 310019, China; 3Fair Friend Institute of Intelligent Manufacturing, Hangzhou Vocational & Technical College, Hangzhou 310018, China; 4State Key Laboratory of Silicon Materials, School of Materials Science and Engineering, Zhejiang University, Hangzhou 310027, China

**Keywords:** lithium-ion batteries, anode, MnS, CoS

## Abstract

MnS has been explored as an anode material for lithium-ion batteries due to its high theoretical capacity, but low electronic conductivity and severe volume change induce low reversible capacity and poor cycling performance. In this work, the nanocapsule consisting of MnS nanopolyhedrons confined in independent, closed and conductive hollow polyhedral nanospheres is prepared by embedding MnCO_3_ nanopolyhedrons into ZIF-67, followed by coating of RF resin and gaseous sulfurization/carbonization. Benefiting from the unique nanocapsule structure, especially inner CoS/C shell and outer pure C shell, the MnS@CoS/C@C composite as anode material presents excellent cycling performance (674 mAh g^−1^ at 1 A g^−1^ after 300 cycles; 481 mAh g^−1^ at 5 A g^−1^ after 300 cycles) and superior rate capability (1133.3 and 650.6 mAh g^−1^ at 0.1 and 4 A g^−1^), compared to the control materials (MnS and MnS@CoS/C) and other MnS composites. Kinetics measurements further reveal a high proportion of the capacitive effect and low reaction impedance of MnS@CoS/C@C. SEM and TEM observation on the cycled electrode confirms superior structural stability of MnS@CoS/C@C during long-term cycles. Excellent lithium storage performance and the convenient synthesis strategy demonstrates that the MnS@CoS/C@C nanocapsule is a promising high-performance anode material.

## 1. Introduction

With the increasing demands of consumable electronics and pure/hybrid electric vehicles and other energy applications, the requirement for advanced energy storage devices, such as lithium-ion batteries (LIBs), becomes higher and higher. It is well-known that anode materials play a significant role in LIBs [1,2,3,4]. However, the traditional graphite material is limited by its low theoretical capacity. Therefore, it is highly urgent to explore advanced anode materials with superior properties. Among the alternative anodic materials, transition metal sulfides (TMSs) have attracted wide interest due to the improved safety, satisfactory theoretical capacity, higher conductivity and lower voltage hysteresis than their metal oxides counterparts [5,6,7,8,9]. In addition, the weak metal–sulfur bond can induce faster reaction kinetics [10,11]. Among TMSs, manganese sulfide (MnS) has the advantages of being ecofriendly, low cost and has a high theoretical capacity (616 mAh g^−1^). Especially, its redox potential is lower than those of other TMSs [12,13,14,15,16,17]. In spite of these advantages, MnS is still subjected to common problems of TMSs. During the charge/discharge process, the volume variation of MnS is large, which results in a serious pulverization problem and poor cycling stability [18,19]. In addition, the low electronic conductivity and sluggish Li^+^ mobility also gives rise to an unsatisfactory rate capability [20,21,22,23,24]. These issues limit the application of MnS as anode material in LIBs.

Many strategies have been proposed to overcome the above shortcomings of MnS. One of the most effective strategies is to combine a wide variety of MnS nanostructures with conductive materials, such as various carbonous materials [25,26,27,28,29,30,31]. On the one hand, the well-designed nanostructures can not only alleviate the volumetric change of MnS, but also shorten diffusion paths of Li^+^ in MnS. On the other hand, hybridizing MnS with carbonous materials can effectively improve the electrical conductivity of MnS, and even suppress volumetric changes of MnS to a certain extent [32,33]. The related works have been extensively reported in recent years. For example, X.J. Xu et al. in-situ synthesized MnS hollow microspheres on reduced graphene oxide sheets. The composite could deliver a reversible capacity of 640 mAh g^−1^ after 400 cycles at 1 A g^−1^ [34]. S. Gao et al. reported flexible MnS–carbon fiber hybrids, and the reversible capacity of 257 mAh g^−1^ was delivered after 1000 cycles at 1 A g^−1^ [26]. L. Zhu et al. prepared MnS/Co_9_S_8_/carbon heterostructures through the chemical vapor deposition treatment of flower-like Mn/Co-glycerate and achieved a high reversible capacity of 608 mAh g^−1^ after 300 cycles at 500 mA g^−1^ [27]. These research findings demonstrate that the composite structure is a key factor for electrochemical performance of MnS [35,36]. It has been found that in most of the composite structures, MnS is either exposed outside carbon materials, or tightly wrapped by various carbon materials without any gap. For the former, it is easy for MnS to fall off from the composite during the discharge–charge process. For the latter, the contact between MnS and the electrolyte is greatly limited. Compared with the above two kinds of composite structures, their intermediate structures are seldom explored. If MnS is locked in an independent, closed and conductive microspace, shortcomings of the above two composite structures can be effectively overcome. Therefore, the innovative research of the composite structure of MnS is still highly desired.

Inspired by the above considerations, herein, we design the independent, closed and conductive hollow polyhedral nanospheres composed of two layers of heterogeneous shells, a CoS/C inner shell and a pure C outer shell. Several MnS nanopolyhedrons are in-situ grown inside the hollow polyhedral nanospheres, constructing a unique nanocapsule structure. The composite is denoted as MnS@CoS/C@C. Compared with other composite strategies, the hollow polyhedral nanospheres can not only improve electronic conductivity of MnS, but also avoid MnS from aggregating or falling off from the composite. More importantly, it can also effectively buffer volume change of MnS. Herein, the detailed structural features and the electrochemical properties of MnS@CoS/C@C nanocapsules have been systematically explored. As expected, MnS@CoS/C@C exhibits superior Li storage performance in terms of reversible capacity, cycling performance and rate capability. The unique nanocapsule structure is also expected to be applied in other fields.

## 2. Results and Discussion

The synthetic procedure of MnS@CoS/C@C nanocapsule is schematically illustrated in Figure 1. First, MnCO_3_ nanopolyhedrons are prepared via a hydrothermal co-precipitation reaction. Under the hydrothermal conditions, citric acid acts as a reducing agent, and reduces Mn^7+^ in KMnO_4_ to Mn^2+^. On the other hand, citric acid is decomposed, and carbonate ions are released. Mn^2+^ combines with CO_3_^2−^. Accordingly, MnCO_3_ is formed. The relevant reactions are shown as follows:KMnO_4_ + C_6_H_8_O_7_ → Mn^2+^ + CO_3_^2−^ + H_2_O
Mn^2+^ + CO_3_^2−^ → MnCO_3_ ↓

Afterword, via the unique bridge effect of PVP between MnCO_3_ and ZIF-67, several MnCO_3_ nanopolyhedrons are assembled into ZIF-67. Furthermore, RF resin is coated on the surface of MnCO_3_@ZIF-67 through a polymerization reaction between resorcinol and formaldehyde. Finally, a gaseous sulfurization and carbonization procedure are performed. MnCO_3_ nanopolyhedrons are sulfurized into MnS nanopolyhedrons. The solid ZIF-67 is sulfurized into the hollow inner shell composed of CoS nanoparticles and a small amount of amorphous carbon. RF resin is decomposed into the outer shell composed of pure amorphous carbon. The above structural changes are accomplished by one-step calcination.

Figure 2a,b show SEM and TEM images of MnCO_3_ nanopolyhedrons. MnCO_3_ nanopolyhedrons present solid structure, uniform size and sharp edges. Although the size is only about 30 nm, they do not aggregate, which ensures the successful embedding of MnCO_3_ nanopolyhedrons into ZIF-67. Through controlling the reaction conditions, MnCO_3_ is restricted to an inadequate growth state, that is, an intermediate morphology close to the cube. Figure 2c exhibits the SEM image of MnCO_3_@ZIF-67. The product presents the typical rhombic dodecahedron, suggesting good crystallinity of ZIF-67. The size of ZIF-67 is also uniform, about 250–450 nm. It can be seen that there is very little scattered MnCO_3_ nanopolyhedrons outside ZIF-67. The TEM image (Figure 2d) reveals that there is several MnCO_3_ nanopolyhedrons inside a ZIF-67 dodecahedron. When MnCO_3_@ZIF-67 is directly sulfurized, the sulfurized product is denoted as MnS@CoS/C. SEM image (Figure 2e) indicates that most of MnS@CoS/C is intact. The surface is smooth. TEM image (Figure 2f) reveals that the sulfurized particles transform from the solid embedded structure into a hollow core-shell structure, resembling a nanocapsule. The thickness of the shell is about 30–40 nm. Inside the hollow sulfurized particles, there exists some more larger solid particles, and they are MnS. Figure 2g shows a local magnified TEM image of the shell. It is found that the shell is composed of CoS nanoparticles. The size of CoS nanoparticles is very small, about 4 nm. The edges of CoS nanoparticles and the interior of the shell are blurred, which is attributed to the existence of a small amount of amorphous carbon.

Figure 3a shows the SEM image of MnCO_3_@ZIF-67@RF resin. Since ZIF-67 may solve in water, the particles transform from the rhombic dodecahedron into smoother polyhedron. The size also becomes smaller. All particles are covered by a layer of uniform smooth resin. Even some particles are stuck together by the resin. This confirms successful coating of RF resin. TEM image (Figure 3b) reveals their internal microstructure. RF resin is outside the particles. Inside the particles, there are some MnCO_3_ that still maintain the initial nanopolyhedron morphology. Among MnCO_3_ nanopolyhedrons, between MnCO_3_ nanopolyhedrons and the RF resin shell, there is no gap. Although most of ZIF-67 is dissolved, MnCO_3_ nanopolyhedrons are still embedded in ZIF-67. Figure 3c shows a SEM image of MnS@CoS/C@C. After the sulfurization and carbonization, the outer RF resin shell is decomposed to a smoother amorphous carbon shell. The TEM mage (Figure 3d) verifies that the internal structure becomes hollow, and MnS nanopolyhedrons exist inside. The composite also becomes a nanocapsule-like core-shell structure. However, in the inside of the shell, many nanoparticles exist. They are CoS nanoparticles formed by sulfurization of ZIF-67. In the outside of the shell, CoS nanoparticles are few, and there is mainly the amorphous carbon wall formed by the carbonization of RF resin. Figure 3d clearly demonstrates that the shell of MnS@CoS/C@C is a two-layer structure. The square box area in Figure 3d is further observed by HRTEM. In Figure 3e, some lattice fringes show d-spacing of 0.301 nm, consistent with the (111) crystal plane of MnS. At the same time, other lattice fringes show d-spacing of 0.292 nm, matching well with the (100) crystal plane of CoS. In addition, the amorphous structure of carbon can be also distinguished simultaneously.

The phases of MnS@CoS/C@C are examined through XRD (Figure 4a). All the characteristic peaks can be assigned to (111), (200), (220), (311), (222) and (400) crystal planes of the cubic phase α-MnS (JCPDS No. 88-2223) and (100), (101), (102) and (110) crystal planes of CoS (JCPDS 75-0605). Diffraction peaks of MnS are relatively strong, which demonstrates high crystallinity and the slightly large size of MnS. On the contrary, diffraction peaks of CoS are very weak, which is consistent with low crystallinity and a very small size of CoS. α-MnS has a cubic rock-salt crystal structure and superior thermal stability, and is the best phase of MnS for lithium storage. There are no other detectable peaks corresponding to ZIF-67, MnCO_3_, RF resin, S and other impurities. This suggests complete sulfurization and carbonization of MnCO_3_@ZIF-67@RF resin. Raman spectrum is conducted to elucidate the degree of graphitization of MnS@CoS/C@C (Figure 4b). At 1355 and 1553 cm^−1^, the two significant Raman scattering peaks appear, known as the D band (the breathing vibration of sp^3^-type carbon) and G band (the stretching modes of sp^2^-type carbon). This demonstrates that RF resin is successfully decomposed to carbon. The intensity ratio of D band and G band (ID/IG) is 0.89, confirming that RF resin-derived carbon is amorphous. Additionally, the two peaks located at 642 and 347 cm^−1^ can be indexed to MnS [26]. The other two weak peaks located at 472 and 517 cm^−1^ are related to E_g_ and F_2g_ peaks of CoS [37]. XRD and Raman results reveal the presence of MnS, CoS and amorphous carbon in the composite.

Figure 4c shows the EDS pattern of MnS@CoS/C@C. In the composite, there are only four elements: Mn, Co, S and C. The atomic ratio of Mn and Co is 4.26. To determine the contents of MnS and CoS in the composite, the TGA of MnS@CoS/C@C is measured under flowing air. As shown in Figure 4d, from the room temperature to 170 °C, a slight weight loss occurs, which is associated with evaporation of adsorbed water in the composite. From 170 °C to 390 °C, an obvious weight increase appears, which is attributed to oxidation of MnS and CoS to MnSO_4_ and CoSO_4_. The weight loss in 390–540 °C is attributed to the combustion of carbon in air. The second weight loss in 615–860 °C is more significant. It is related to the decomposition of MnSO_4_ and CoSO_4_ into Mn_2_O_3_ and Co_3_O_4_. Based on the atomic ratio of Mn and Co provided by EDS, the contents of MnS, CoS and carbon are calculated to be 60%, 14.61% and 25.39%, respectively.

The electrochemical properties of MnS@CoS/C@C as anode material for LIBs are studied in coin-type half-cell. Figure 5a shows CV curves of MnS@CoS/C@C at a scan rate of 0.1 mV s^−1^ in the first three cycles. CV curves present a lot of reduction peaks and oxidation peaks from MnS and CoS, indicating that both MnS and CoS participate in electrochemical reactions, and the reactions of MnS and CoS are multi-step. The first cathodic scanning curve exhibits five reduction peaks located at 1.63, 1.20, 0.58, 0.25 and 0.01 V, corresponding to the intercalation of Li^+^ into MnS, conversion of CoS to metallic Co, conversion of Li_2_MnS into metallic Mn and Li_2_S, the formation of the SEI film and intercalation of Li^+^ into carbon, respectively. The subsequent anodic scanning curves are relatively stable, and show seven oxidation peaks located at 0.25, 1.12, 1.28, 1.90, 2.06, 2.27 and 2.35 V, related to the deintercalation of Li^+^ from carbon (0.25 V), re-conversion of metallic Mn to Li_2_MnS (1.12 V), re-conversion of metallic Co to Li_x_CoS (1.28 V), deintercalation of Li^+^ from Li_2_MnS (1.9 and 2.35 V) and deintercalation of Li^+^ from Li_2_CoS (2.06 and 2.27 V), respectively [17,38,39]. The subsequent cathodic scanning curves also become stable, and six peaks are presented at 1.92, 1.61, 1.31, 0.76, 0.35 and 0.01 V. The cathodic peaks at 1.61, 0.76 and 0.33 V are associated with intercalation and conversion reactions of MnS. The cathodic peaks at 1.92 and 1.31 V correspond to intercalation and conversion reactions of CoS. Except for the first scanning, changes in CV curves are negligible, indicating the excellent structure stability and electrochemical reversibility of MnS@CoS/C@C.

To further elucidate the electrochemical behavior of MnS@CoS/C@C, the charge/discharge curves at the current density of 1 A g^−1^ at the 1st, 2nd and 5th cycle are investigated. The control material of MnS is prepared by the sulfurization of MnCO_3_ nanopolyhedrons. The control material of CoS/C is prepared by sulfurization of ZIF-67. The charge/discharge curves (Figure 5b,c) of MnS and CoS exhibit multiple potential plateaus. These plateaus well coincide with oxidation peaks and reduction peaks of CV curves of MnS@CoS/C@C, which confirms that the electrochemical reactions of MnS@CoS/C@C are composed of those of MnS and CoS. This point can be further confirmed by the charge/discharge curves of MnS@CoS/C@C (Figure 5d). It is found that potential plateaus of MnS@CoS/C@C basically correspond to oxidation and reduction peaks of CV curves of MnS@CoS/C@C, meanwhile coincide with potential plateaus of the control materials of MnS and CoS. Furthermore, compared with MnS and CoS, the charge/discharge curves of MnS@CoS/C@C almost overlap, indicating the better stability and reversibility. It is worth noting that the initial capacity loss of MnS@CoS/C@C is lower than those of the control materials of MnS and CoS/C, which is attributed to the RF resin-derived carbon shell that greatly improves the initial Coulomb efficiency [20].

Figure 6a shows rate performance of MnS@CoS/C@C nanocapsule. At the current densities of 0.1, 0.2, 0.5, 1, 2, 4 A g^−1^, MnS@CoS/C@C delivers average discharge capacities of 1133.3, 1082.6, 1039.6, 952, 823.4, 650.6 mAh g^−1^. The discharge capacities at different current densities are extraordinary, which suggests excellent rate capability of MnS@CoS/C@C. When the current comes back to 1 and 0.1 A g^−1^, the discharge capacity can recover to 890 and 1244 mAh g^−1^. This demonstrates outstanding stability of MnS@CoS/C@C. For comparison, rate performance of MnS@CoS/C is also provided in Figure 6a. Its average discharge capacities stabilize at 814, 755, 671, 605, 509, 333 mAh g^−1^ at 0.1, 0.2, 0.5, 1, 2, 4 A g^−1^. It is found that discharge capacity of MnS@CoS/C is 300–400 mAh g^−1^ lower than that of MnS@CoS/C@C. This huge difference demonstrates significance of the outer pure carbon shell. It greatly improves electronic conductivity and structural stability of MnS@CoS/C. Figure 6b compares rate performance of MnS@CoS/C@C and other advanced MnS and CoS composites. MnS@CoS/C@C is superior to many previous reported composites, demonstrating its structural superiority. Figure 6c displays charge/discharge curves of MnS@CoS/C@C at different current densities. As the current increases from 0.1 to 1.0 A g^−1^, the potential plateaus of the charge/discharge curves shift very slightly. Only at 2 and 4 A g^−1^ does the overpotential become slightly greater. This further exhibits high electronic conductivity, low polarization and excellent charge–discharge ability of MnS@CoS/C@C.

Figure 6d compares cycling performance of MnS@CoS/C@C and the control materials of MnS@CoS/C and MnS at 1 A g^−1^ over 300 cycles. MnS prepared by direct sulfurization of MnCO_3_ nanopolyhedron is not protected by any substance, and the conductivity is not improved. As a result, its initial discharge capacity is 713 mAh g^−1^. After 3 cycles, the discharge capacity rapidly drops to 225 mAh g^−1^. Afterwards, it maintains a stable, but low capacity. At the 300th cycle, the discharge capacity is only 118 mAh g^−1^. When MnS is protected by the CoS/C hollow body, the initial capacity increases to 1200 mAh g^−1^. Nevertheless, after 50 cycles, the capacity of MnS@CoS/C becomes like MnS. This is because the CoS/C shell easily pulverizes during cycling, and cannot provide adequate structural support for the inner MnS core. As for MnS@CoS/C@C, its discharge capacity is several times those of MnS and MnS@CoS/C. At the 74th cycle, the discharge capacity reaches 1033 mAh g^−1^. At the 300th cycle, it still delivers a discharge capacity of 674 mAh g^−1^. This comparison clearly reveals the importance of the inner CoS/C shell and the outer pure C shell, especially the outer pure C shell, for improving the cycling performance of MnS. The outer pure C shell remarkably enhances the electronic conductivity and structural stability of the composite. The cycling performance of MnS@CoS/C@C is further explored at a higher current of 5 A g^−1^ (Figure 6e). The discharge capacity of MnS@CoS/C@C slowly decreases from 673 mAh g^−1^ at the 1st cycle to 481 mAh g^−1^ at the 300th cycle, with a capacity retention ratio of 71.5%. The cyclability at 5 A g^−1^ is also impressive. Table 1 compares cycling performance of MnS@CoS/C@C and other advanced MnS composites, demonstrating that MnS@CoS/C@C has excellent cycling performance in terms of reversible capacity and stability.

The reaction kinetics of MnS@CoS/C@C is further investigated by CV curves at various scan rates in order to find out the reason for the high specific capacity and excellent rate capability. In Figure 7a, as scan rate increases from 0.1 to 1.0 mV s^−1^, CV curves always keep the similar shapes, and all oxidation and reduction peaks shift very slightly. This suggests that the polarization of MnS@CoS/C@C is very small. Three distinct peaks are selected to analyze the relationship between peak current (*i*) and scan rate (*v*), according to the equation (*i* = *av^b^*). Herein, *a* and *b* are the variable parameters. It is known that the value of *b* can be used to estimate the type of the reaction kinetics. In general, a *b*-value of 0.5 means that diffusion completely controls the electrochemical reaction, whereas a *b*-value of 1.0 represents that the capacitive effect dominates the electrochemical reaction. *a* and *b* can be determined by the plot of log *i* vs. log *v*, as shown in Figure 7b. The three groups of peaks present good linear correlation. The *b* value of the cathodic peak (Peak 1) is fitted as 0.79. The *b* values of the two anodic peaks (Peak 2 and 3) are fitted as 0.74 and 0.80. The values suggest that the pseudocapacitive effect controls electrochemical reaction of MnS@CoS/C@C. To quantitatively elucidate the contribution ratios of the pseudocapacitive effect and the diffusive effect in the electrochemical reaction, another equation (*i* = *k*_1_*v* + *k*_2_*v*^1/2^) is used to separate the current response (*i*) of CV curves. In the equation, *k*_1_*v* indicates the pseudocapacitive current, and *k*_2_*v*^1/2^ represents the diffusive current. *k*_1_ and *k*_2_ are two parameters related to the potential and can be determined by plotting *i/v*^1/2^ vs. *v*^1/2^ and then performing linearly fitting. *k*_1_ and *k*_2_ are the slope and the intercept of the fitting line. In this way, the CV curve at scan rate of 1.0 mV s^−1^ can be decomposed to the pseudocapacitive current and the diffusive current, as shown in Figure 7c. It can be seen that the pseudocapacitive contribution accounts for 88.75% of the total charge storage (highlighted by green). At other scan rates of 0.1, 0.2, 0.4 and 0.8 mV s^−1^, the pseudocapacitive contributions are 63.06, 67.56, 74.53 and 83.31%, respectively, as illustrated in Figure 7d. The results confirm that the pseudocapacitive effect is dominant. The significant pseudocapacitive contribution is mainly attributed to the nanocapsule structure. In the nanocapsule, the sizes of MnS and CoS are very small, and they do not aggregate. The abundant inner void exists in the nanocapsule, facilitating electrolyte infiltration and ionic transfer. The above factors remarkably enhance reaction kinetics, and improve rate performance of MnS@CoS/C@C.

The reaction kinetics of MnS@CoS/C@C are further studied by EIS. Figure 7e exhibits the Nyquist plots before cycling and after the 5th cycle. The equivalent circuit is also inserted. The inclined straight line in the low-frequency region corresponds to the lithium-ion diffusion process inside the electrode material, represented by the Warburg impedance (Z_w_). In the high and medium frequency regions, the compressed semicircles are associated with the interfacial SEI film resistance (R_i_) and charge transfer resistance (R_ct_). R_Ω_ represents internal resistance. CPE is a constant phase element. At the 5th cycle, the semicircle becomes smaller, which means that R_ct_ decreases. The fitted results indicate that R_ct_ decreases from 64.6 Ω before cycling to 38.2 Ω at the 5th cycle. The low R_ct_ suggests rapid charge transfer and a rapid interface reaction. R_Ω_ is only 5.7 Ω, which indicates that the outer pure carbon shell and the inner CoS/C shell improve the electronic conductivity of the composite. EIS results demonstrate that the reaction kinetics of MnS@CoS/C@C are fast, which is beneficial to obtain high capacity and good rate performance.

To ascertain structural stability of MnS@CoS/C@C during long-term charge/discharge cycles, the MnS@CoS/C@C electrode after 300 cycles at 5 A g^−1^ is observed. SEM image (Figure 8a) shows that there are numerous spherical particles on the electrode. These particles are intact. The TEM image (Figure 8b) reveals that MnS nanoparticles do not pulverize and are still well confined inside the nanocapsules. Colloid-like polymer fills the inner void of the nanocapsules. Overall, the nanocapsule structure of MnS@CoS/C@C is well maintained. This convincingly verifies that MnS@CoS/C@C is extremely stable. The excellent structural stability is attributed to the two shells. The inner CoS/C shell increases the capacity of the composite, promotes ion transport and enhances the structural strength of the composite. For the outer pure C shell, its role is more important. It significantly enhances the structure strength and electronic conductivity of the composite. Furthermore, the internal abundant void space can well buffer volume variation of MnS and the two shells, and meanwhile store electrolyte for electrochemical reaction.

## 3. Experimental

### 3.1. Material Synthesis

Synthesis of MnCO_3_ nanopolyhedrons: 0.2 g KMnO_4_ and 0.2 g citric acid were dissolved in 40 mL deionized water, and then stirred for 4 h at the ambient temperature. The mixed solution was transferred to a 50 mL Teflon-lined stainless-steel autoclave, and then heated to 160 °C for 4 h. The precipitate was collected via centrifugation, and then washed with ethanol several times. The product was dried in an oven overnight at 50 °C.

Synthesis of MnCO_3_@zeolitic imidazolate framework (ZIF)-67: 0.55 g MnCO_3_ nanopolyhedrons were ultrasonically dispersed in 50 mL methanol. 1.1 g polyvinylpyrrolidone (PVP, M_w_ = 58,000) was added under stirring to prepare solution A. Solution B was prepared by dissolving 2.96 g 2-methylimidazole in 50 mL methanol. Solution C was prepared by dissolving 0.20 g Co(NO_3_)_2_·6H_2_O in 25 mL methanol. The solution B and solution C were poured into solution A under stirring. After reaction for 45 min, the mixed solution was aged at room temperature for 12 h. The dark purple precipitate was separated by centrifugation, washed repeatedly with methanol, and dried overnight at 50 °C in vacuum.

Synthesis of MnCO_3_@ZIF-67@resorcinol-formaldehyde (RF) resin: Solution D was prepared by ultrasonically dispersing 0.35 g MnCO_3_@ZIF-67 in 12.5 mL ethanol. Solution E was prepared by dissolving 0.23 g cetyltrimenthylammonium bromide (CTAB) in 30 mL deionized water. Solution E was poured into solution D under stirring. After 15 min, 0.12 g resorcinol and 0.2 mL ammonium hydroxide were added to the mixed solution. After 20 min, 0.017 mL formaldehyde solution was added. After reaction for 12 h at room temperature, the product was separated, washed with ethanol 3 times, and finally dried.

Synthesis of MnS@CoS/C@C: MnCO_3_@ZIF-67@RF resin and sulfur powder were placed in a horizontal quartz tube furnace, with a weight ratio of 1:6. The sulfur powder was at upstream. The mixture was then annealed at 600 °C for 2 h with a heating rate of 2 °C min^−1^ under flowing argon with a flowing rate of 150 sccm.

### 3.2. Materials Characterizations

Scanning electron microscopy (SEM, Ultra-55) and transmission electron microscope (TEM, JEOL JEM-2100) were utilized to observe morphology and microstructure. Information about the crystal structure were obtained by Rigaku D/max2550VL/PC system X-ray diffractometer (XRD) with Cu-Kα radiation. The chemical composition was determined by INCA energy dispersive X-Ray spectroscopy (EDS) equipped in SEM. Raman spectra were collected from DXR Raman microspectrometer with a 633 nm laser. Thermogravimetric analysis (TGA) curve was collected with a thermogravimetric analyzer (NETZSCH TG209F3) under air flow at a heating ramp of 10 °C min^−1^.

### 3.3. Electrochemical Measurements

The working electrodes were made by mixing active material (70 wt%), carbon black (Super-P) (20 wt%) and polyvinylidene fluoride (PVDF) (10 wt%) to form a homogeneous slurry. N-methylpyrrolidone was used as solvent. The slurry was then spread onto copper foils with a diameter of 1.2 cm, followed with drying in vacuum at 70 °C overnight to remove NMP. The CR2025 coin-type half-cells were assembled in an Ar-filled glove-box under the contents of moisture and oxygen less than 0.5 ppm. The polypropylene membrane (Celgard 2400) was used as separator. The electrolyte was the mixture of dimethyl carbonate and ethylene carbonate (1:1 in volume) dissolved with 1 M LiPF_6_. Li foil was used as counter electrode. The galvanostatic charge–discharge cycling measurements were operated in the voltage range of 0.01–3.0 V at room temperature on NEWARE CT-3008 battery test systems. The electrochemical impedance spectroscopy (EIS) with a frequency range of 0.01 HZ—100 KHz and the cyclic voltammetry (CV) curves over the potential range from 0.01 to 3.0 V (vs. Li^+^/Li) were tested on VersaSTAT 3 electrochemical measurement system. After a certain cycling numbers, the MnS@CoS/C@C electrode was taken out from the cell, rinsed with dimethyl carbonate, and then dried in vacuum for SEM and TEM observation.

## 4. Conclusions

In summary, we developed a facile in-situ assembling and gaseous sulfurization/carbonization strategy to construct MnS@CoS/C@C, in which several MnS nanopolyhedrons as cores are perfectly confined inside hollow polyhedral nanospheres composed of an inner CoS/C shell and outer pure C shell. As an anode material for LIBs, the MnS@CoS/C@C electrode exhibits high specific capacity (1133.3 mAh g^−1^ at 0.1 A g^−1^), improved cycling stability (481 mAh g^−1^ at 5 A g^−1^ after 300 cycles) and excellent rate capability, as well as rapid reaction kinetics, including a high proportion of pseudocapacitive effect and low reaction impedance. The improved electrochemical performance is associated with the unique nanocapsule structure. Two layers of shells, especially outer pure C shell, not only increase electronic conductivity, but also improve structural stability. Abundant void space in the nanocapsule also facilitates electrolyte storage. The synthesis strategy and structure features proposed in this work can motivate innovative exploration of novel hybrid composites for high-performance energy storage applications.

## Figures and Tables

**Figure 1 molecules-28-00898-f001:**
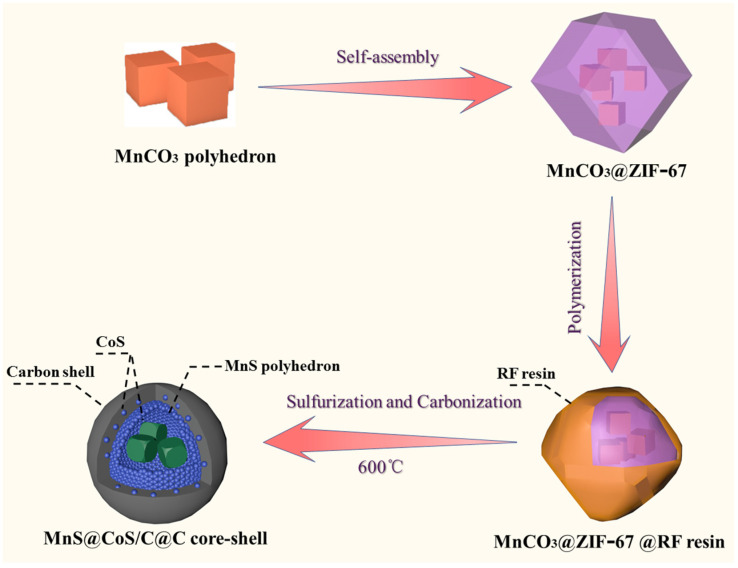
Synthetic procedure for the preparation of MnS@CoS/C@C nanocapsules.

**Figure 2 molecules-28-00898-f002:**
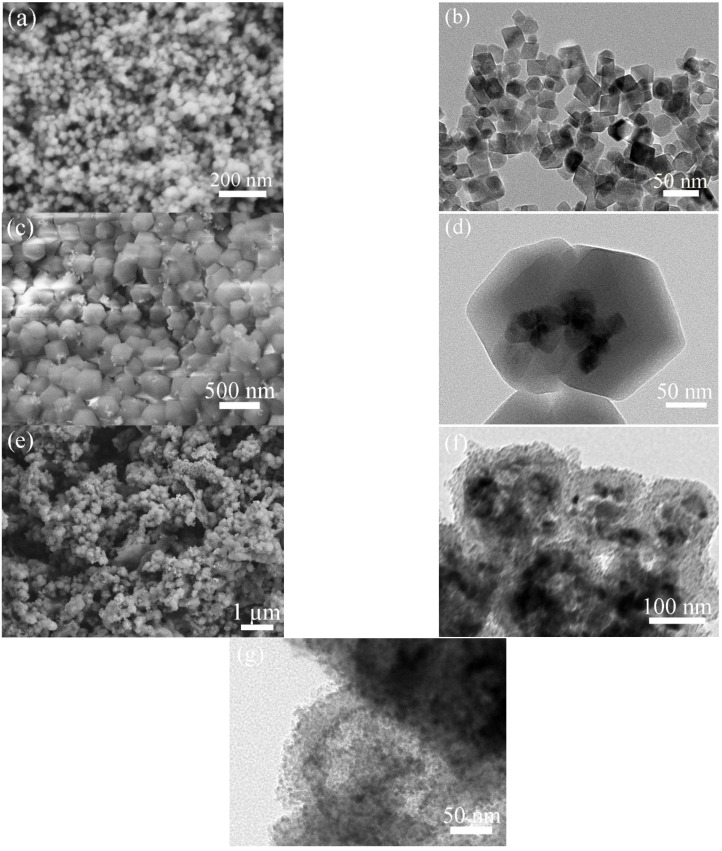
(**a**) SEM and (**b**) TEM images of MnCO_3_ nanopolyhedrons; (**c**) SEM and (**d**) TEM images of MnCO_3_@ZIF-67; and (**e**) SEM, (**f**) TEM and (**g**) local TEM images of MnS@CoS/C.

**Figure 3 molecules-28-00898-f003:**
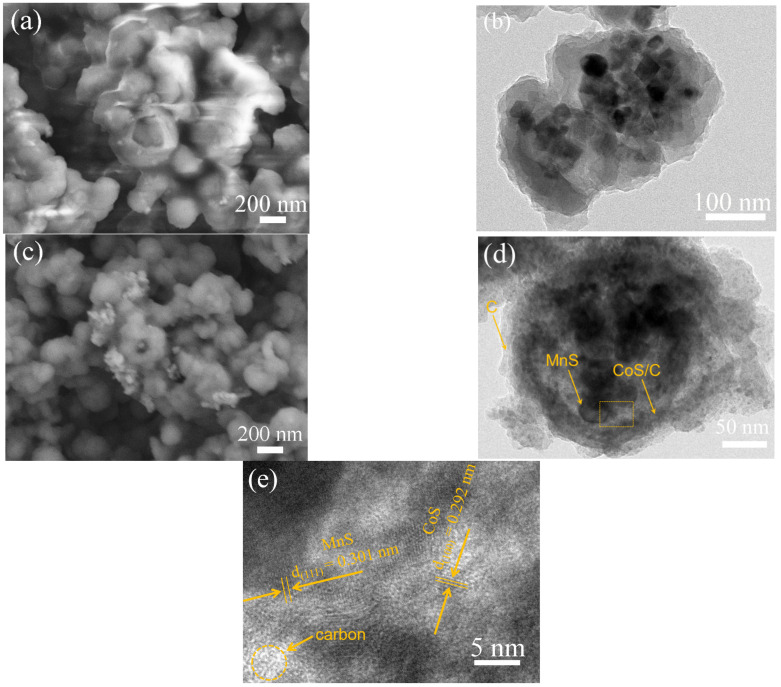
(**a**) SEM and (**b**) TEM images of MnCO_3_@ZIF-67@RF resin; and (**c**) SEM, (**d**) TEM and (**e**) HRTEM images of MnS@CoS/C@C.

**Figure 4 molecules-28-00898-f004:**
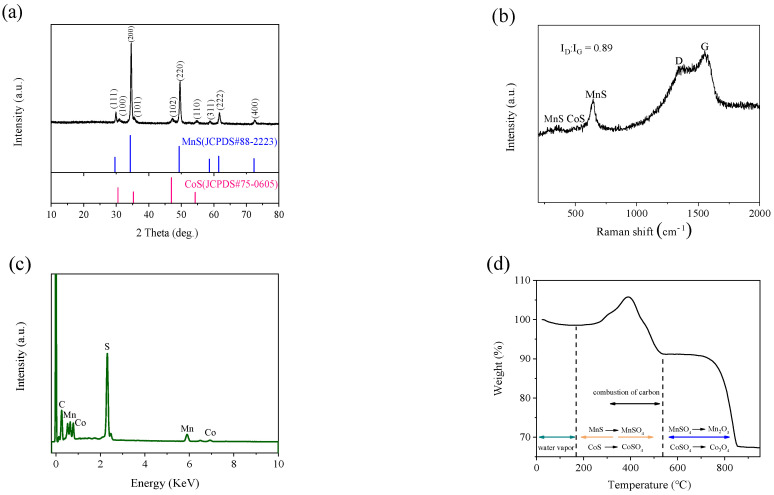
(**a**) XRD pattern, (**b**) Raman spectrum, (**c**) EDS pattern and (**d**) TGA curve in air of MnS@CoS/C@C.

**Figure 5 molecules-28-00898-f005:**
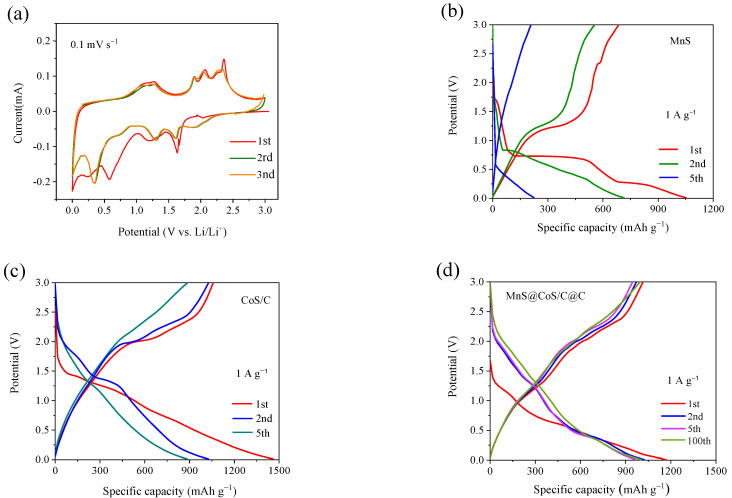
(**a**) CV curves of MnS@CoS/C@C at the scan rate of 0.1 mV s^−1^ in the first three cycles, Galvanostatic charge–discharge curves of (**b**) MnS, (**c**) CoS/C, (**d**) MnS@CoS/C@C at current density of 1 A g^−1^.

**Figure 6 molecules-28-00898-f006:**
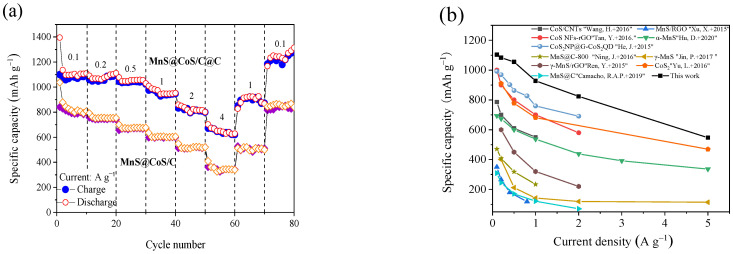
(**a**) Rate performance of MnS@CoS/C@C and MnS@CoS/C, (**b**) Comparison of rate capability of MnS@CoS/C@C and α-MnS [21], MnS@C-800 [25], MnS/RGO [34], CoS/CNTs [40], CoS NFs-rGO [41], γ-MnS [42], MnS@C [43], γ-MnS/rGO [44], CoS_2_ [45], CoS_2_ NP@G-CoS_2_QD [46], (**c**) Galvanostatic charge–discharge curves of MnS@CoS/C@C at various current densities, (**d**) Longer-term cyclic performances and coulombic efficiency of MnS@CoS/C@C and the control materials of MnS@CoS/C and MnS at 1 A g^−1^ and (**e**) Longer-term cyclic performances and coulombic efficiency of MnS@CoS/C@C at 5 A g^−1^.

**Figure 7 molecules-28-00898-f007:**
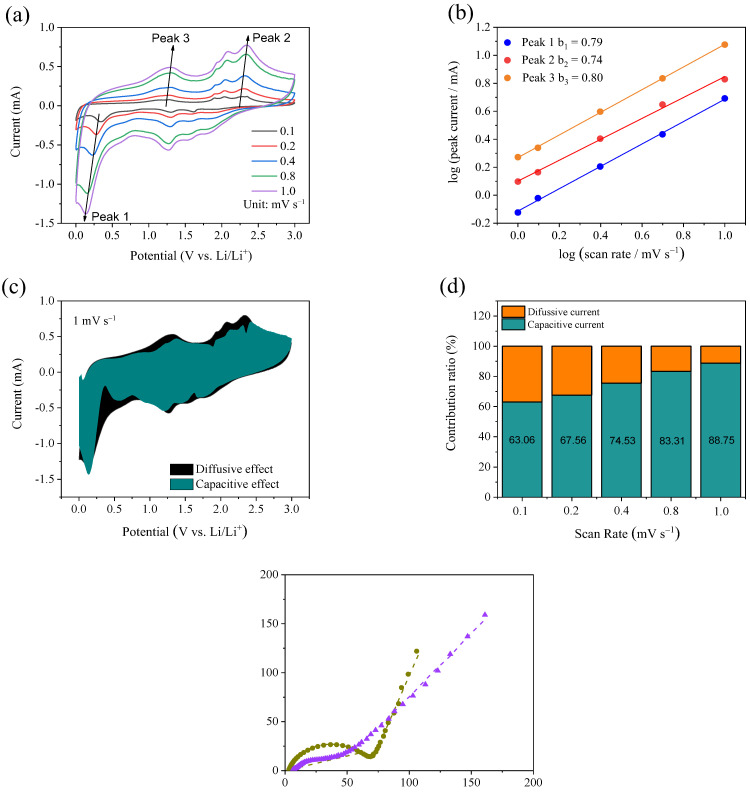
(**a**) CV curves of MnS@CoS/C@C at scan rates of 0.1, 0.2, 0.4, 0.8 and 1.0 mV s^−1^, (**b**) linear fitting of log (*i*) vs. log (*v*) of three oxidation/reduction peaks, (**c**) capacitive contribution (green region) and diffusive contribution (black region) to charge storage at scan rate of 1.0 mV s^−1^, (**d**) the normalized contribution ratios of the capacitive effect at different scan rates and (**e**) Nyquist plots and the fitting curves before cycling and after 5 cycles, with the equivalent circuit model.

**Figure 8 molecules-28-00898-f008:**
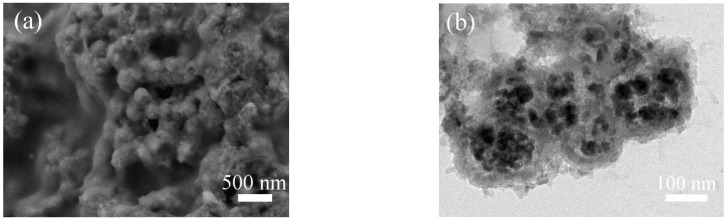
(**a**) SEM and (**b**) TEM images of MnS@CoS/C@C after 300 cycles at 5 A g^−1^.

**Table 1 molecules-28-00898-t001:** Cycling performance comparison between MnS@CoS/C@C and other reported MnS, CoS composites.

Materials	Current Density (A g^−1^)	Specific Capacity (mAh g^−1^)	Cycling Performance	Ref.
CoS/Graphene	0.06	749	40	[39]
CoS/CNTs	0.3	465	100	[40]
CoS NFs-rGO	0.1	939	100	[41]
MnS/RGO	1	640	400	[34]
α-MnS	0.1	870	200	[21]
CoS_2_-MnS@rGO	0.1	1324	100	[20]
γ-MnS	0.2	350	300	[42]
MnS@C	1	578	100	[43]
γ-MnS/rGO	0.2	600	100	[44]
MnS@C-800	0.5	200	100	[25]
CoS_2_	1	737	200	[45]
CoS_2_ NP@G-CoS_2_ QD	1	831	300	[46]
MnS@CoS/C@C	1	1001	100	This work
	1	674	300	
	5	481	300	

## Data Availability

The data presented in this study are available on request from the corresponding authors.

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
