# Peer review of "Nanocapsule of MnS Nanopolyhedron Core@CoS Nanoparticle/Carbon Shell@Pure Carbon Shell as Anode Material for High-Performance Lithium Storage"

_molecules, 2023, doi:10.3390/molecules28020898_

Round 1

Reviewer 1 Report

Excellent lithium storage performance and the convenient synthesis strategy demonstrate that the MnS@CoS/C@C nanocapsule is a promising high-perfor- mance anode material. The manuscript should be accepted after minor revision.

How about the working potential of as-prepared anode compared to the graphite anode ?

The full cell evaluation or calculation should be included in this paper to make sure the exactly high energy density of as-prepared anode.

Reviewer 2 Report

Referee Report

on paper “ Nanocapsule of MnS nanopolyhedron core@CoS nanoparticle/carbon shell@pure carbon shell as anode material for high-performance lithium storage “ (molecules-2138069) by authors P. Yang, Y.F. Yuan, D. Zhang, Q.H. Yang, S.Y. Guo and J.P. Cheng submitted to Molecules

This is interesting paper. It reports the preparation and investigation of the structure and electrochemical performance of the nanocapsule composite consisting of MnS nanopolyhedrons confined in independent, closed and conductive hollow polyhedral nanospheres through embedding MnCO3 nanopolyhedrons into ZIF-67, followed by coating of RF resin and gaseous sulfurization/carbonization. Benefiting from the unique nanocapsule structure, especially inner CoS/C shell and outer pure C shell, the MnS@CoS/C@C composite as anode material presents excellent cycling performance (674 mAh g-1 at 1 A g-1 after 300 cycles and 481 mAh g-1 at 5 A g-1 after 300 cycles) and superior rate capability (1133.3 and 650.6 mAh g-1 at 0.1 and 4 A g-1), better than the control material of MnS, MnS@CoS/C, and other MnS composites. The presented data are reliable and useful. However, paper needs some improvement only after implementation of which it can be published:

1.    The authors should mention in 1. Introduction some information about the doped sulfides obtained earlier:

(1). S.V. Trukhanov, I.V. Bodnar, M.A. Zhafar, Magnetic and electrical properties of (FeIn2S4)1−x(CuIn5S8)x solid solutions, J. Magn. Magn. Mater. 379 (2015) 22-27. https://doi.org/10.1016/j.jmmm.2014.10.120.

(2). I.V. Bodnar, M.A. Jaafar, S.A. Pauliukavets, S.V. Trukhanov, I.A. Victorov, Growth, optical, magnetic and electrical properties of CuFe2.33In9.67S17.33 single crystal, Mater. Res. Express 2 (2015) 085901. https://doi.org/10.1088/2053-1591/2/8/085901.

2.    I understand the choice of object of study. These are the doped sulfides based nancmpostes which have excellent electronic properties. I fully agree with the authors that: “ Among the alternative anodic materials, transition metal sulfides (TMSs) have attracted wide interest due to the improved safety, high theoretical capacity, higher conductivity and lower voltage hysteresis than metal oxides counterparts [5-9]. “. However, the alloys compounds are not free from some disadvantages. One of which is their low resistance to the aggressive influence of environmental factors such as temperature, oxygen and electromagnetic radiation. The oxide compounds in this sense are much more stable when used up to 1000 ºC. With that, there are different classes of oxide materials with excellent electronic properties. One of them is the complex iron oxides called as the ferrites:

(3). A.V. Trukhanov, V.O. Turchenko, I.A. Bobrikov, S.V. Trukhanov, I.S. Kazakevich, A.M. Balagurov, Crystal structure and magnetic properties of the BaFe12−xAlxO19 (x=0.1–1.2) solid solutions, J. Magn. Magn. Mater. 393 (2015) 253-259. https://doi.org/10.1016/j.jmmm.2015.05.076.

(4). M.V. Zdorovets, A.L. Kozlovskiy, D.I. Shlimas, D.B. Borgekov, Phase transformations in FeCo – Fe2CoO4/Co3O4-spinel nanostructures as a result of thermal annealing and their practical application, J. Mater. Sci.: Mater. Electron. 32 (2021) 16694-16705. https://doi.org/10.1007/s10854-021-06226-5.

This information should be noted in 1. Introduction.

3.    In 3. Experimental it is necessary to indicate of another new synthesis methods of functional transition metal materials:

(5). A.L. Kozlovskiy, M.V. Zdorovets, Synthesis, structural, strength and corrosion properties of thin films of the type CuX (X = Bi, Mg, Ni), J. Mater. Sci.: Mater. Electron. 30 (2019) 11819-11832. https://doi.org/10.1007/s10854-019-01556-x.

(6). T.I. Zubar, T.I. Usovich, D.I. Tishkevich, O.D. Kanafyev, V.A. Fedkin, A.N. Kotelnikova, M.I. Panasyuk, A.S. Kurochka, A.V. Nuriev, A.M. Idris, M.U. Khandaker, S. V. Trukhanov, V.M. Fedosyuk, A.V. Trukhanov, Features of galvanostatic electrodeposition of NiFe films with composition gradient: influence of substrate characteristics, Nanomaterials 12 (2022) 2926. https://doi.org/10.3390/nano12172926.

4.    It is well known that the combination of different compounds which have excellent electronic properties leads to new composite materials which have earned great technological interest in recent years. The addition of a second phase can significantly improve the electronic properties of the resulting composite material:

(7). A.L. Kozlovskiy, M.V. Zdorovets, Effect of doping of Ce4+/3+ on optical, strength and shielding properties of (0.5-x)TeO2-0.25MoO-0.25Bi2O3-xCeO2 glasses, Mater. Chem. Phys. 263 (2021) 124444. https://doi.org/10.1016/j.matchemphys.2021.124444.

(8). M.A. Almessiere, N.A. Algarou, Y. Slimani, A. Sadaqat, A. Baykal, A. Manikandan, S.V. Trukhanov, A.V. Trukhanov, I. Ercan, Investigation of exchange coupling and microwave properties of hard/soft (SrNi0.02Zr0.01Fe11.96O19)/(CoFe2O4)x nanocomposites, Mat. Today Nano 18 (2022) 100186. https://doi.org/10.1016/j.mtnano.2022.100186.

This issue should be mentioned and discussed in 1. Introduction and 2. Results and discussion.

5.    The presented 8 papers should be inserted in References.

The paper should be sent to me for the second analysis after the major revisions.

Reviewer 3 Report

This manuscript reports a core and double shell MnS composite anode material with enhanced conductivity and Li ion transport kinetics. Benefitted from the new structure, a stable cycling performance of 674 mAh g−1 can be achieved at 1 Ag−1. This manuscript is recommended for publication if the following comments can be addressed.

1.     Please specify the role of CoS in for example structural stability or ion transport, if a control sample MnS@C is not provided. 

2.     It should be specified that the specific capacity is normalized to MnS only or the whole composite, since MnS accounts for only 60 wt% and the capacity calculated in this work is high.

3.     The initial capacity loss is of MnS@CoS/C@C is reduced compared with the control sample in Figure 5. Since MnS, CoS and carbon usually has low first cycle efficiency, can the author explain the improvement? 

4.      

Round 2

Reviewer 2 Report

The paper has been well corrected and it can be recommended.